# Analysis and Suppression Design of Stray Light Pollution in a Spectral Imager Loaded on a Polar-Orbiting Satellite

**DOI:** 10.3390/s23177625

**Published:** 2023-09-02

**Authors:** Shuaishuai Chen, Xinhua Niu

**Affiliations:** 1Shanghai Institute of Technical Physics, Chinese Academy of Sciences, Shanghai 200083, China; chenshuaishuai@mail.sitp.ac.cn; 2Key Laboratory of Infrared Detecting & Imaging Technology, Chinese Academy of Science, Shanghai 200083, China; 3University of Chinese Academy of Sciences, Beijing 100049, China

**Keywords:** polar-orbiting meteorological satellite, MERSI, stray light radiation model, PST, stray light test

## Abstract

As the non-imaging light of optical instruments, stray light has an important impact on normal imaging and data quantification applications. The FY-3D Medium Resolution Spectral Imager (MERSI) operates in a sun-synchronous orbit, with a scanning field of view of 110° and a surface imaging width of more than 2300 km, which can complete two coverage observations of global targets per day with high detection efficiency. According to the characteristics of the operating orbit and large-angle scanning imaging of MERSI, a stray light radiation model of the polar-orbiting spectrometer is constructed, and the design requirements of stray light suppression are proposed. Using the point source transmittance (PST) as the merit function of the stray light analysis method, the instrument was simulated with all stray light suppression optical paths, and the effectiveness of stray light elimination measures was verified using the stray light test. In this paper, the full-link method of “orbital stray light radiation model-system, internal and external simulation design-system analysis and actual test comparison verification” is proposed, and there is a maximum decrease in the system’s PST by about 10 times after applying the stray light suppression’s optimization design, which can provide a general method for stray light suppression designs for polar-orbit spectral imagers.

## 1. Introduction

Stray light refers to the light transmitted abnormally in the optical system, including light leakage, the residual reflection of optical surfaces, the residual reflection of non-optical surfaces (such as the inner wall of the lens barrel), and scattered light caused by optical surface quality problems [1]. With continuous improvements in the requirements for the quantitative application of satellite remote sensing data, the influence of in-orbit stray light on the image quality and quantitative accuracy of the imager has received more and more attention [2]. The in-orbit data show that the effects of stray light include the effects on remote sensing imaging and onboard calibration. For example, the AVHRR spectral imager loaded on the NOAA satellite has abnormal signals in the medium- and long-wave infrared channels when the satellite enters and exits the shadow area of Earth to calibrate the onboard blackbody, and the problem is mainly caused by the sun’s pollution with respect to the onboard blackbody [3]. The solar diffuser of the Terra satellite spectral imager MODIS is affected by the strong scattering of sunlight due to the Earth’s atmosphere, resulting in short-term fluctuations in the calibration coefficients of some channels [4]. The remote sensing images of the NPP satellite’s VIIRS imager low-light channel are also contaminated with solar stray light in the satellite’s day–night region [5].

In terms of stray light analyses and suppression methods, Zhao [6] analyzed the stray light pollution of the camera by incident sunlight outside the orbit and gave the path and influence of the incident sun. Jiang [7] carried out stray light analyses and provided suppression designs using the terahertz optical system and obtained stray light distributions in different bands. Clermont [8] and Wei [9] carried out stray light design work using large off-axis three-mirror anastigmatic telescopes and verified the effectiveness of the design. Lee [10] used the instrument’s calibrated field-of-view image to complete the stray light correction of the VIIRS low-light channel’s image. Ando [11] used the moon’s image to characterize and correct the instrument’s stray light. Several algorithms have been developed to control stray light using post-processing based on the calibration of stray light properties, such as Clermont [12], Laherrere [13], and Zong [14]. It can be observed that the analysis and suppression of stray light mainly focus on the design of the stray light suppression of the instrument itself and the analysis and reprocessing of stray light characteristics with respect to imaging images.

The Fengyun-3 meteorological satellite (FY-3) is a satellite operating in the polar sun-synchronous orbit, and its Medium-Resolution Spectral Imager (MERSI) uses a scanning mirror to scan Earth at a scanning width of 2300 km, which can complete two global observations per day with high detection efficiencies. The instrument’s detection band covers a range from 0.4 μm to 12.5 μm, and the quantization level is 12 bit, with the dynamic range reaching 100% solar albedo in visible bands, which is mainly used for global, all-weather, multispectral quantitative detection. Remote sensing data can be used to achieve medium-term numerical forecasting and to monitor natural disasters and ecological environments in order to study the laws of global climate change and environmental change [15].

Aiming at the application of high-sensitivity detection and the high quantification of polar-orbiting meteorological satellite spectral imagers, this paper carries out research on stray light analysis and suppression technology based on orbital characteristics, and we establish an orbital stray light radiation response model, adopt a combination of external and internal stray light analysis and suppression methods, and propose corresponding stray light suppression measures. At last, the instrument’s stray light suppression ability is verified via the stray light test.

## 2. Methods

This paper mainly starts from the external and internal stray light sources of the imager and carries out stray light simulation analyses and tests on the entire optical path; the specific process is shown in Figure 1.

The current stray light sources for polar-orbiting satellite spectral imagers can be divided into two categories:

1. Stray light outside the orbit, which means stray light outside the target imaging (Earth) area, including the sun’s irradiation of the satellite body, the scattering and reflection of sunlight by mechanical components around the scanning mirror, etc.

By selecting the sunlight vector of the satellite, the angle between sunlight and the imager’s flight direction (satellite’s *X*-axis), the imager’s subsatellite point direction (satellite’s *Z*-axis), and the imager’s cold sky direction (satellite’s *Y*-axis) is established to obtain the orbital sunlight illumination of the imager in orbit throughout the year.

2. Stray light outside the field of view within the scanning range, which refers to the stray light caused by the incident light outside the field of view during the imaging process of the imager, including the bright reflection of the Earth’s atmospheric clouds and solar flares in the Earth’s oceans.

The analysis of stray light in the scanning field of view is mainly the analysis of the external non-imaging light suppression ability of the imager’s optomechanical system, and full optical path ray tracing is mainly carried out using Monte Carlo’s method. Here, it is necessary to complete the setting of the imager’s optomechanical system model, including the setting of optical path parts, mechanical surface characteristics, and tracing ray settings. The photomechanical material properties and surface properties of the imager should represent material properties in the most accurate manner possible, and the test conditions should correspond to the actual imaging situation of the imager. The tracing ray settings should be able to converge, which means that the energy reaching the focal plane does not differ by orders of magnitude when increasing or decreasing the number of rays.

On the basis of completing the simulation analysis of the imager, the special stray light test system is used to test the stray light of the imager in the entire optical path. The band range is selected, the coverage angle and light source intensity are tested, the system’s transmittance before and after the imager’s optimization is analyzed, and the test results are compared with the simulation results to verify the effectiveness of the simulation analysis.

### 2.1. Stray Light Analysis and Suppression Methods

The stray light analysis method comprises the critical surface and illuminated surface analysis methods summarized by Robert P. Breault [16]. The critical surface is the surface that the detector can see, including the surface formed by the lens that can be observed by the detector, and these surfaces are the source of all scattered light. The illuminated surface is the surface of the optical system that is illuminated by an external light source. The important surface is the intersection of the critical surface and the illuminated surface. All stray light paths involve scattering on the important surface, which means that it is better when there are fewer important surfaces with respect to the system. Distinguishing critical, illuminated, and important surfaces in a system is a common method for analyzing stray light in the industry, which is further used to analyze stray light characteristics in optomechanical systems.

Stray light suppression methods include blocking and removal methods, blocking the stray light path from the illuminated surface to the critical surface, and removing the illuminated surface and the critical surface. In stray light suppression measures, the setting of the optical stop is fundamental. The field stop directly reduces the illuminated surface, and the Lyot stop directly restricts the critical surface. The field stop is placed as far from the entrance as possible, and the effective aperture stop is placed as far from the detector as possible. In secondary imaging systems, the field stop and the Lyot stop are paired to achieve good occlusion. Scattering and diffraction from the aperture stop are blocked at the Lyot stop, and the use of multi-cascading field stops and Lyot stops can further reduce diffracted stray light.

### 2.2. Optical System Stray Light Test Method

The influence of stray light on the imaging system can be evaluated using the system’s point source transmittance (PST) function. PST is defined as the ratio of the irradiance, *Ed*(*θ*), generated by the point light source at angle *θ* after passing through the optical system to the irradiance, *Ei*(*θ*), perpendicular to the point light source at the entrance port of the imager, and its formula is described as follows:(1)$$PST=\frac{Ed(\theta )}{Ei(\theta )}$$

The point source transmittance ratio, PST, reflects the attenuation ability of the optical system itself relative to the point stray light source. The smaller the PST, the stronger the system’s stray light suppression ability. By testing different off-axis angle PSTs, the image’s surface response curves with different off-axis angles can be given, and the stray light suppression magnitude of the optical system can be obtained.

## 3. Simulation Analysis and Suppression Design of Stray Light in MERSI

### 3.1. Analysis and Suppression of Stray Light Outside the Orbit

From the current satellite’s in-orbit data, it can be observed that direct sunlight on the satellite’s mounting surface is one of the key paths for stray light in the spectral imager. Firstly, the incidence angle distribution of sunlight is obtained according to the analysis of orbital parameters. Secondly, the layout of the shaper plate is reasonably set according to the optical path structure in order to protect the key components of the instrument from direct sunlight.

The orbital parameters of the FY-3D satellite are used to obtain orbital simulation, and the orbital parameters are shown in Table 1.

The condition for the polar orbit satellite spectral imager illuminated by sunlight in orbit is as follows: the angle between the sunlight vector and the *Z*-axis of the satellite coordinate system is greater than 90°. At this time, sunlight can obliquely shine on the load installation surface of the satellite, which is above the instrument. The illumination period of each orbit is the stage during which the satellite enters and exits the Earth’s shadow area. Geographically, it is located near the Earth’s north and south poles, as shown in Figure 2. The red color in the figure represents the solar vector, and the yellow color represents the satellite’s coordinate system.

The annual variation range of the angle between the solar vector and the *Y* axis of FY-3D is between 105° and 123°, the angle between the solar vector and the satellite XOZ plane(θ_y_) is between 15° and 33°, the angle between the satellite XOY plane (θ_z_) is between 0° and 29°, and the annual distribution of the angle between the solar vector and the satellite Y axis at the moment when the satellite enters and exits the Earth’s shadow is shown in Figure 3.

These incidence parameters can be used to model solar illuminance and the stray illuminance of the sun $\left(E_{SunStr}\right)$ incident on the focal plane of the instrument:(2)$$E_{SunStr}=E_{Sun}(\alpha ,\beta )\cos\alpha Pst(\alpha ,\beta )$$
where *α* is the zenith angle of the solar incidence, and *β* is the azimuth of the solar incidence.

The focal plane signal of the Earth’s target, $\left(E_{Signal}\right)$, in the imager’s field of view is described as follows:(3)$$E_{Signal}=\frac{L_{Earth}\tau \pi }{4F^{2}}$$
where *τ* is the transmittance of the instrument, and F is the F# of the imager’s optical system.

Considering the Earth’s occlusion of sunlight, the maximum incidence zenith angle of sunlight is 62°. Using Formulas (2) and (3), the stray light of the sunlight can be estimated: when the PST of the instrument outside the axis’s range of 62° is better than 3.4 × 10^−4^, the influence of solar stray light on normal imaging is less than 1%.

According to the annual solar incidence of the satellite, the layout of the payload platform, and the surrounding environment, the Fengyun-3D star imager effectively blocks these direct rays by adding shades on both sides of the +Y axis of the scanning mirror, and the size of the shield needs to meet the following condition: W > Ltanθ_y_, $H>H_{s}+\sqrt{L^{2}+W^{2}}tan\theta_{z}$.

Here, L is the maximum length of the scanning mirror, Hs is the maximum height of the scanning mirror, and θy and θz are the maximum values of the year. The sunshade’s design is shown in Figure 4.

According to the ground test and on-orbit verification of FY-3D MERSI, in the area near the North and South Poles, the sunshade can block the sun’s irradiation on the scanning mirror and blackbody calibration area well, and the calibration data of the medium- and long-wave infrared channels during this period are normal, which means that the sunshade can effectively protect the scanning mirror and calibration blackbody from light pollution from the sun [17], and the data value fluctuation caused by stray light is less than two digital numbers (DNs). The full-scale digital output of the imager is 4096.

### 3.2. Simulation Analysis and Suppression of Stray Light Inside the Spectral Imager

In order to achieve an imaging range of 2300 km at an altitude of 836 km, MERSI uses a 45° mirror to scan and carry out imaging processes, and it has a scanning range of over 110°. The influence of stray light outside the imaging field of view within a wide scanning range cannot be ignored. The schematic diagram of the instrument’s scanning angle is shown in Figure 5.

Therefore, the stray light analysis and suppression technology of the imager are studied with respect to both external and internal aspects. On the one hand, the stray light pollution of the instrument caused by external light such as the sun is reduced. On the other hand, the instrument’s ability to suppress stray light can be improved using a reasonable internal design.

According to orbital altitudes, by imaging the field of view and other parameters, the imager’s in-orbit stray light energy model is established, which includes the following.

The stray illumination of the Earth target ($E_{EarthStr}$) incident to the focal plane outside the field of view of the instrument is described as follows:(4)$$E_{EarthStr}=\iint L_{Earth}(\theta ,\varphi )\Omega (\theta ,\varphi )Pst(\theta ,\varphi )d\theta d\varphi$$

Earth’s target radiance, $L_{Earth}(\theta ,\varphi )$, is estimated as follows:(5)$$L_{Earth}(\theta ,\varphi )=E_{Sun}/\pi \cdot \cos(\theta )\cos\varphi e^{\tau_{0}/\cos\theta \cos\varphi }$$

The stereoscopic angle $\Omega \left(\theta ,\varphi \right)$ of the imaging space of the Earth target to the imager is
(6)$$\Omega (\theta ,\varphi )=\frac{R^{2}[(R+h)\cos\theta \cos\varphi -R]}{[R^{2}+(R+h{)}^{2}-2R(R+h)\cos\theta \cos\varphi {]}^{3/2}}$$
where *θ* is the longitude of the Earth’s target, *φ* the latitude of the Earth’s target, *τ*_0_ is the atmospheric transmittance, *R* is the radius of the Earth, *h* is the satellite altitude, and Pst is the instrument’s transmittance distribution.

These models can be used to calculate and analyze the stray light of MERSI loaded on the polar-orbiting FY-3D satellite, and the orbital height is 836 km, the orbital inclination angle is 98.75°, and the orbital local time of ascending node is 14:00~14:30. Using Formulas (3) and (4), the stray light of the Earth can be estimated: when the average PST of the instrument in the off-axis range of 62° is better than 6.4 × 10^−4^, the influence of stray light on the normal imaging of the Earth is less than 1%.

The optical design of MERSI is shown in Figure 6, including the coaxial RC telescope, the fold mirror, the anti-image spinning K-mirror, the spectroscopic assembly, and the rear optical mirror set. The main parameters of the optical system are shown in Table 2.

The optical system contains two focal planes in the visible/near-infrared band, which adopts normal temperature working detectors. The focal plane of the short-wave infrared ~long-wave infrared band adopts a low-temperature refrigeration detector, which works inside the radiation cooler. Stray light analysis is mainly for the simulation analysis and testing of visible/near-infrared focal planes. The model diagram of the MERSI spectral imager is shown in Figure 7 using a 45° scanning and imaging mirror. The target passes through the main optics (co-axial RC telescope), fold mirror (to twist light path), and K mirror (to eliminate image rotation) and enters the spectroscopic assembly and aft optics to form spectral detection channels in various bands. In addition to reducing the area of direct external light on the scanning mirror, it is necessary to further improve the elimination capacity inside the instrument to reduce the incidence of non-imaging light that is reflected and scattered onto the detection focal plane by other mechanical surfaces around the imager.

We set the BSDF properties of bidirectional scattering distributions on each surface of the system, and the BSDF model based on ABg [18] is used to set the parameters of different surfaces:(7)$$BSDF=\frac{A}{B+{\left|\vec{\beta }-\vec{\beta_{0}}\right|}^{g}}$$
where $\vec{\beta }$ is the projection of the scattering direction unit vector on the surface, $\vec{\beta_{0}}$ is the projection of the reflection direction unit vector on the surface, A is the maximum scattering amount, and B and g reflect the concentration of scattering.

The ABg model is used to model the BSDF properties of the surfaces. The surface of the mechanical parts is modeled as light absorption at 0.9, specular reflectance is 1 × 10^−5^, transmittance is 0, Lambert scattering is 0.09999, A is 0.06365, B is 1, and g is 0. The surface of the optical lens is modeled as light absorption at 0.01, specular reflectance is 0.01, specular transmittance is 0.9797, Lambert scattering is 0.00013, the BRDF integral is 1.324 × 10^−4^, parameter A is 1 × 10^−5^, B is 0.015, g is 2, the BTDF integral is 1.4499 × 10^−4^, parameter A is 1 × 10^−5^, B is 0.01, and g is 2. The surface of the color separation sheet needs to be designed using spectroscopy, the reflectivity of the visible light separation sheet is 0.96 for 0.4~0.58 μm, and the transmittance of 0.6~1.05 μm is 0.96 so as to achieve light splitting.

As the first imaging of the optical system, the telescope’s main lens tube needs to be designed with a hood, and this includes the inlet hood design and the internal open hood design of the primary mirror. The telescope’s inlet hood is set to meet the maximum imaging field of view of the telescope, and the safe distance from the scanning mirror is also taken into account. The design is shown in Figure 8, with red light being the telescope’s maximum field of view.

The design of the internal open hood of the primary mirror has met the requirements of the normal imaging field of view’s light incidence, and it is designed according to the effective envelope of imaging light. As shown in Figure 9, the red light denotes effective imaging light, and the black part is the hood.

Ray tracing settings and importance sampling need to be well designed, and this would ensure the effectiveness and repeatability of the simulation. The external grid light source is an annular source that is 600 mm in diameter, and it covers the entrance of the imager. The gird’s orientation uses direction vectors: Its normal vector is (0,0,1), and its up vector is (0,1,0). The irradiance of the grid source is set as 10 × 10^6^ W/m^2^ to obtain weak stray light illumination on the focal plane. Importance sampling can help improve the efficiency of simulation analyses by increasing the incident probability of the sampling area. For example, the inner surface of the primary mirror hood has an importance sampling setting that increases the scattering light in the fold mirror.

On the basis of completing the model’s design, a combination of forward- and reverse-tracing analysis methods is used to obtain the critical and illuminated surfaces of the instrument’s optical system. The detector emits light (reverse tracing) to locate the critical surfaces in the system, which are the surface of the fold mirror’s frame and the inner wall of the primary mirror’s hood. The external grid light source (forward tracing) is used to locate the illuminated surfaces of the system, which are the surfaces of the fold mirror frame, the primary mirror hood, and the inner wall of the main optics hood, as shown in Table 3.

For the suppression of stray light within the system, the main goal is to reduce the number of critical and illuminated surfaces and block the direct path from the critical surface to the illuminated surface. Based on simulation analysis results, the internal optical and mechanical structure of the instrument is optimized, and it mainly includes blocking the transmission path of stray light and adding targeted internal apertures.

The main measures here include the following:Aperture stop setting: In the optical system, the aperture stop limits the three-dimensional angle of the beam that reaches the image surface detector of the system. For objects in object space, except for imaging objects, center occlusion, or field-of-view vignetting, other radiation passing through the aperture stop will not reach the detection image surface of the system. The imager mainly uses the primary mirror as the aperture stop, and the stray light incident outside the field of view can be effectively suppressed by adding the primary lens tube hood.Lyot stop settings: The characteristics of the Lyot stop are the same as that of the aperture stop, which can limit the number and size of critical surfaces of the system in front of the Lyot stop outside the system’s field of view and reduce the transmission path of stray light radiation. A blackened Lyot stop at the fold mirror and K mirror of the imager is set to suppress stray light after the main optics, as shown in Figure 10.Field stop settings: For optical systems with an intermediate image plane, a field stop can be set near the intermediate image plane to effectively suppress the incidence of stray light outside the field of view, as shown in Figure 11.Special treatment of critical surfaces: Further dissipation designs of critical surfaces and illuminated surfaces have been used in the system. On the basis of the original deep blackening treatment, the oxidation method, which forms a rough surface or increases fine lines, is used to reduce the scattering and reflectivity of each surface and achieves a better suppression effect with respect to internal stray light.

### 3.3. PST Testing and Verification of the Imaging System

The dedicated point source transmittance testing system can be used to verify the stray light suppression design of the spectral imager and to obtain the PST curve of the instrument before and after optimization. The point light source transmittance, PST, reflects the attenuation ability of the optical system relative to point stray light sources. The smaller the PST, the stronger the system’s ability to suppress stray light. Via simulation analysis and the testing of PST with different angles, the system’s response curves can be obtained, thereby obtaining the order of magnitude of stray light suppression in the optical system.

The point source transmittance stray light test system is mainly composed of three parts, namely the stray light source simulation system, the high-precision rotating stage, and the low-light detection system. The stray light source simulation system is used to simulate the infinity point source, which comprises a 0.66 μm laser light source and a 1 m aperture off-axis parallel light tube. The high-precision rotating stage provides the two-dimensional movement of the azimuth and pitch angle of the tested optical system to carry out PST tests at different fields of view. The low-light detection system has the ability to detect low-light signals. The entire test system is shown in Figure 12 [19].

The PST test was carried out in a dark room with ambient humidity of 40% and temperatures of 20%, as shown in Figure 13. The workflow of the point source transmittance stray light test system is as follows. The laser light source system gathers the laser beam at the focal plane of the collimator tube and emits the collimated directional light through the collimator to illuminate the pupil of the tested system, and the angle between the collimated light and the optical axis of the test system is θ. After directional light is carried out by the photometer system, it reaches the focal plane via the optical system, mechanical structure surface scattering, or aperture diffraction. The detector system measures the irradiance at the focal plane, and the PST of the photometer system at angle θ can be obtained by comparing it with the illuminance before entering the measured system. The turntable realizes the operation of the azimuth and pitch angle of the tested system to achieve PST measurements of different angles of incidence.

## 4. Result and Discussion

On the basis of completing system modeling and optimizing the design of stray light suppression, the system’s ability to eliminate stray light is analyzed. The simulation and testing experiment mainly complete the PST measure of the entire spectral imager before and after optimization in the X (flight dimension) and Y (scanning dimension) directions using a test angle of ±90°.

Figure 14 shows the PST distribution curve of MERSI as a function of the illumination angle in the X (flight dimension) and Y (scanning dimension) directions. The blue dot indicates the experimental results of the PST before optimization, the red angle indicates the experimental results of the PST after optimization, and the green line indicates the simulation results of the PST after optimization.

After implementing the above measures to eliminate stray light, the stray light incident on the system’s focal plane outside the field of view is well suppressed. In the instrument scanning dimension (scanning mirror scanning direction) and flight dimension (satellite flight direction), the PST is below 6.4 × 10^−4^ (the field of view’s angle is larger than 10°). Simulation analyses show that these measures can effectively reduce the impact of stray light on the system’s imaging outside the imaging field of view.

The comparison of the measured data in the figure shows that there is good consistency between the simulation and test results in terms of numbers and shapes. After using optimization measures to eliminate stray light, the PST level of the imager significantly decreased. There is a maximum decrease of about one order of magnitude in the scanning direction within the incident range beyond 60° and over one order of magnitude in the flight direction within the incident range of 10° to 60°, which means that the system’s ability to suppress stray light has been effectively improved.

As Section 3.1 and Section 3.2 indicate, in order to achieve the goal of stray light pollution that is better than 1%, the PST of the instrument should be better than 3.4 × 10^−4^ outside the axis range of 62°, and the average PST inside the off-axis range of 62° should be better than 6.4 × 10^−4^. From the perspective of PST values, the scanning dimension and flight dimension’s PSTs are better than 1 × 10^−4^ outside ±10°, and the average PST is better than 2 × 10^−4^ inside ±62°, which can meet the system’s requirements for suppressing stray light.

Uncertainties in the system’s PST test include inconsistencies in the measurement instrument, the stability of the light source, and the responsivity of the detector. This experimental measurement is carried out in a confined dark room, and the environmental conditions have negligible errors due to visible illuminance measurements. The measurement’s accuracy analysis is as follows:The light source adopts a 450 mW laser and is equipped with a laser power stabilizer. When it is used and preheated to a stable range, the radiation energy fluctuation of the light source is 0.1~0.2%, and the relative uncertainty of the light source is 0.2%.The uniformity of the 1 m parallel light beam is 85%. Considering that the size of the instrument inlet is 200 mm, the actual non-uniformity is better than 1%.The measurement error of the low-light detector includes the stability of the detector’s spectral response, the temperature of the detector’s dark current, the indication’s uncertainty, etc., and the uncertainty is 2%.

The above content is a single error analysis of the Class B uncertainty of the measuring device, and the uncertainty of the measurement is synthesized according to the synthesis method of measurement uncertainty in error theory, and the final synthetic uncertainty is better than 2.3%.

## 5. Conclusions

In this paper, the full-link stray light analysis method of the polar-orbiting meteorological satellite spectral imager was proposed: Firstly, the idea of establishing an orbital stray light radiation model was adopted to analyze the sources and effects of orbital stray light pollution using the model, and the stray light suppression requirements and optimization directions required by the system were given. Secondly, the simulation analysis method combining external and internal parts was adopted to realize the comprehensive improvement of the external stray light suppression and internal decontamination optimization designs of the imager, and the stray light suppression measures and optimized point source transmittance curve were obtained to meet the requirements of the model. Finally, the stray light test before and after the optimization of the imager was completed by using the special stray light test system, and the transmittance curve of the stray light point source before and after optimization was obtained. The test and simulation analysis curves were compared to verify the rationality and effectiveness of the stray light simulation analysis design of the instrument.

The experimental results show that the full-link stray light analysis method proposed in this paper is reasonable, feasible, and effective. It realizes the stray light simulation and analysis idea compared to the original large number of simulation analyses and shifts to the establishment of the orbital stray light radiation model. The polar-orbit stray light model can be used to guide the direction of simulation analysis and verify the suppression measures by carrying out experiments; these applications form a closed-loop design, which can guide the design and testing of stray light using related imagers.

## Figures and Tables

**Figure 1 sensors-23-07625-f001:**
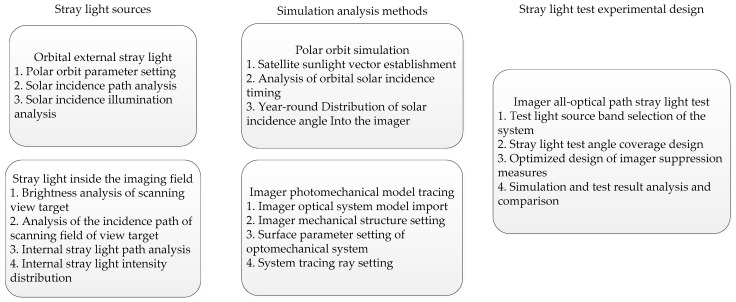
Stray light analysis and testing procedures.

**Figure 2 sensors-23-07625-f002:**
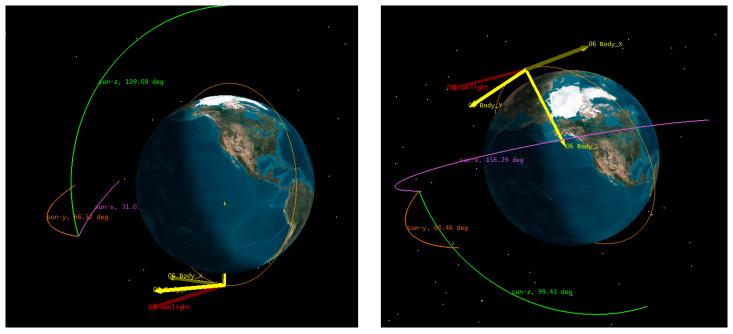
Sunlight incidence area per orbit.

**Figure 3 sensors-23-07625-f003:**
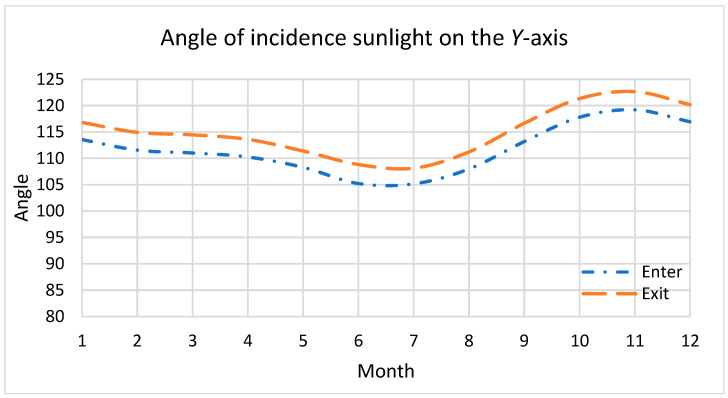
Annual distribution of the angle between the sun and the *Y*-axis of satellites.

**Figure 4 sensors-23-07625-f004:**
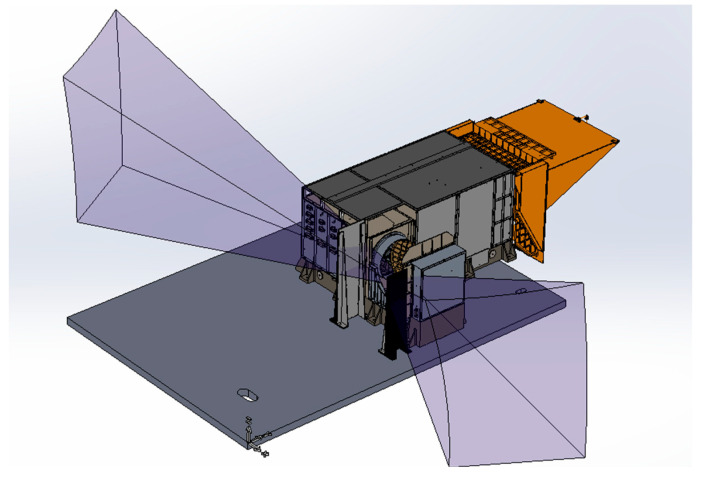
Annual angle distribution and sunshade setting of MERSI.

**Figure 5 sensors-23-07625-f005:**
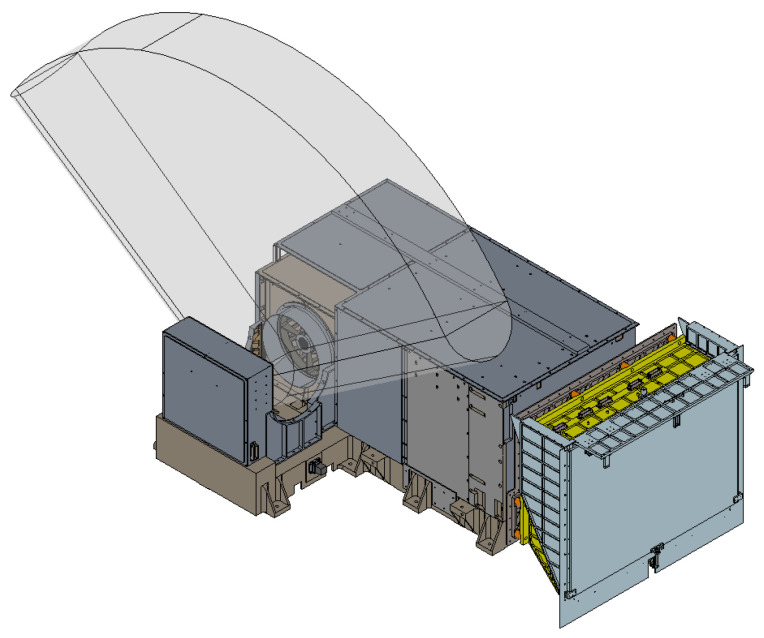
Schematic diagram of MERSI’s scanning angle.

**Figure 6 sensors-23-07625-f006:**
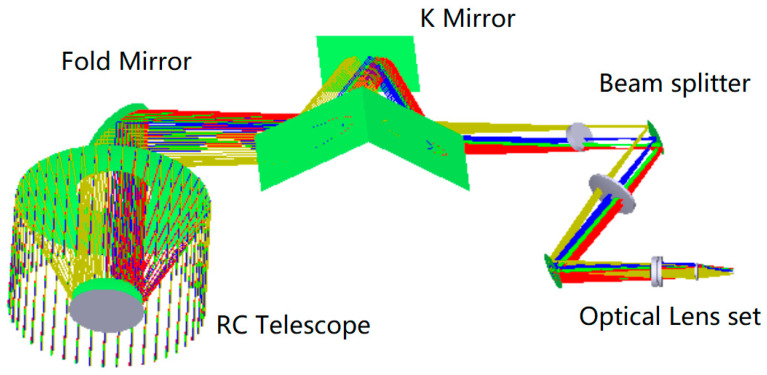
Schematic diagram of the optical path.

**Figure 7 sensors-23-07625-f007:**
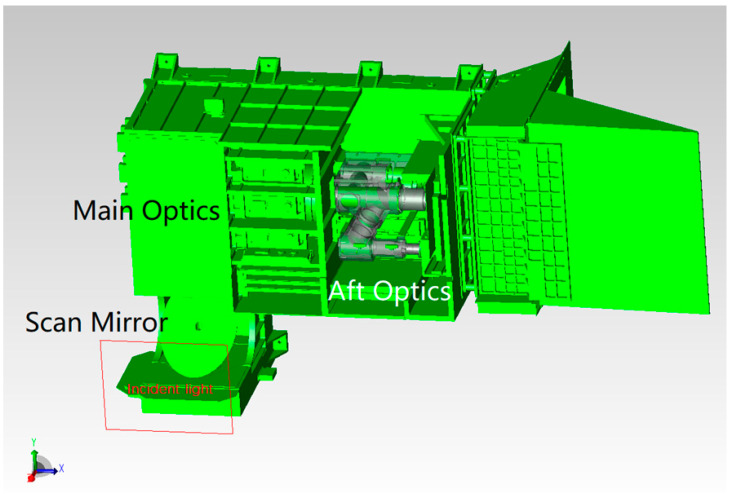
Schematic diagram of the spectral imager.

**Figure 8 sensors-23-07625-f008:**
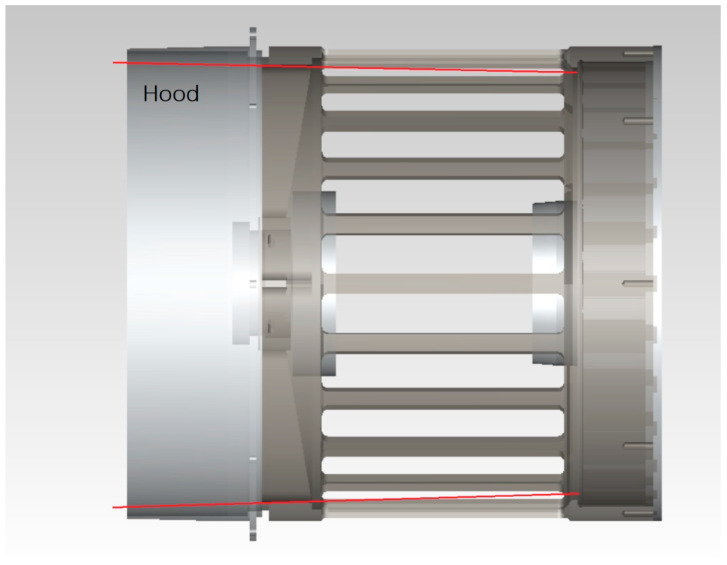
Telescope inlet hood design.

**Figure 9 sensors-23-07625-f009:**
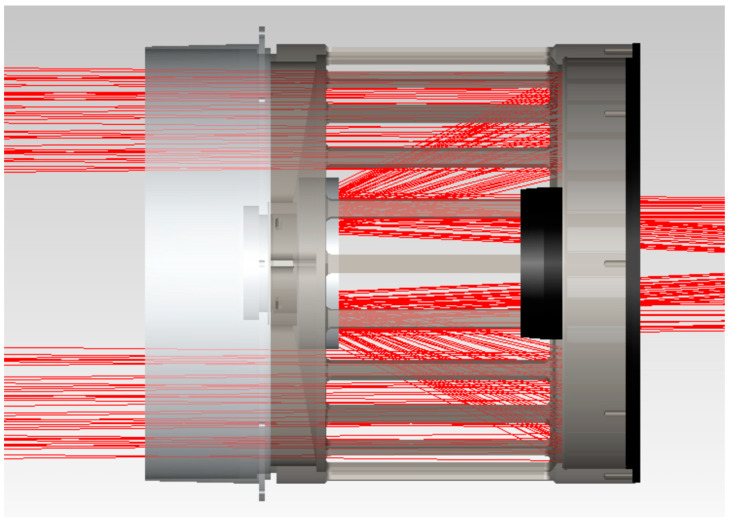
Telescope’s primary mirror hood design.

**Figure 10 sensors-23-07625-f010:**
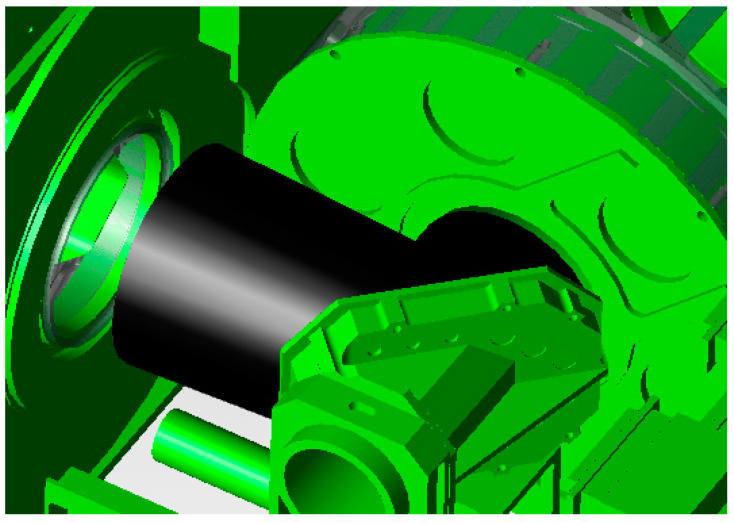
Design of the fold mirror’s aperture.

**Figure 11 sensors-23-07625-f011:**
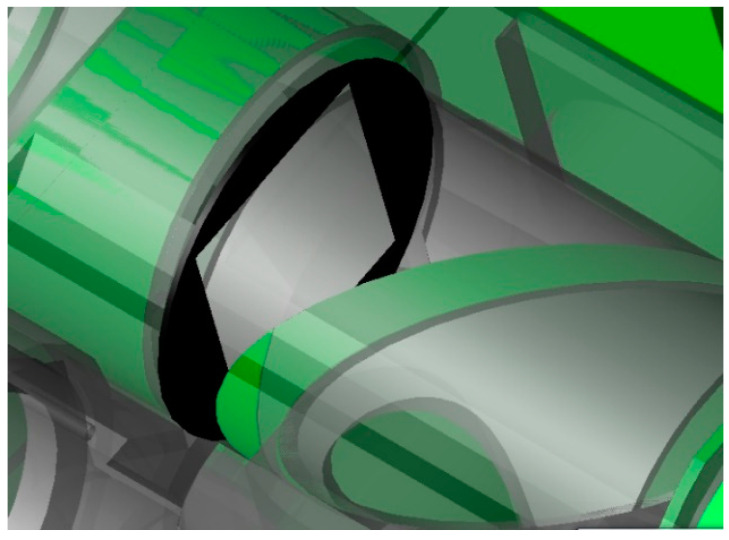
Field stop schematic.

**Figure 12 sensors-23-07625-f012:**
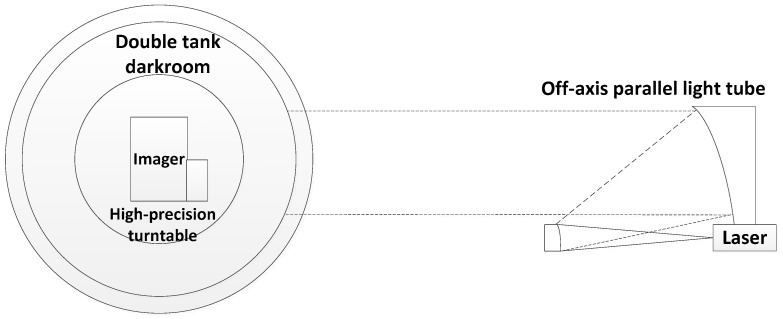
Two-dimensional PST test system.

**Figure 13 sensors-23-07625-f013:**
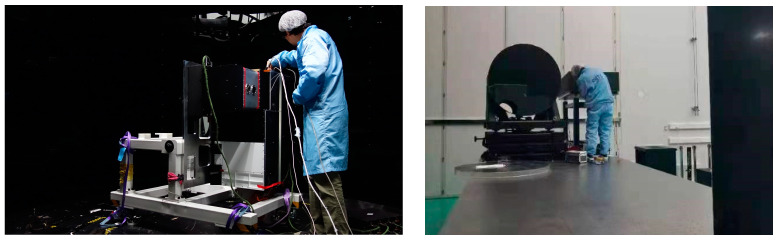
PST test preparation and equipment status.

**Figure 14 sensors-23-07625-f014:**
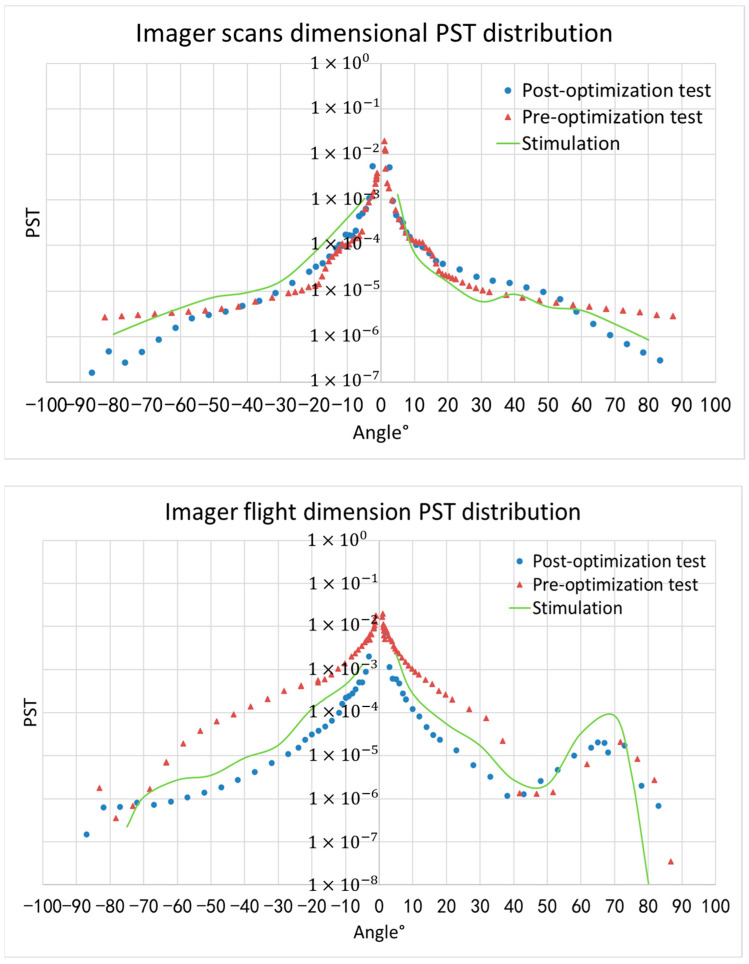
Scan and flight dimension PST distribution curve of MERSI.

**Table 1 sensors-23-07625-t001:** Orbit parameters of FY-3D.

Orbit Type	Near-Polar Sun-Synchronous Orbit
Nominal height	836 km
Inclination	98.75°
Regression period range	4~10 Days
Eccentricity	≤0.0025
Launch window	Ascending local time 13:40~14:00

**Table 2 sensors-23-07625-t002:** System optical parameters.

Optical Aperture (mm)	Detection Band (μm)	Focal Length (mm)	Instantaneous Field of View (mrad)	Nadir Resolution (m)
200	0.40~0.60	500.00	0.3, 1.2	250, 1000
	0.65~1.03	500.00	0.3, 1.2	250, 1000
	1.38~2.13	250.00	1.2	1000
	3.80~8.55	333.33	1.2	1000
	10.8~12.0	333.33	0.3	250

**Table 3 sensors-23-07625-t003:** Critical and illuminated surfaces of the optical system.

	**Fold Mirror Frame**	**Inner Wall of the Primary Mirror Hood**
Critical Surface	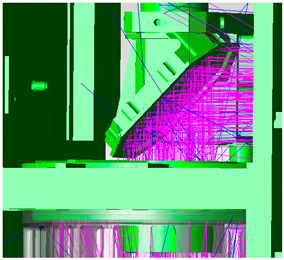	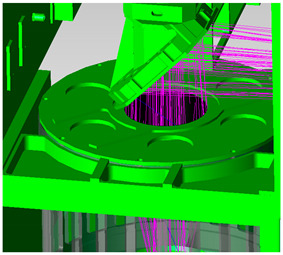
	**Fold Mirror Frame**	**Primary Mirror Hood**
Illuminated surface	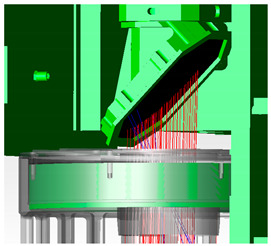	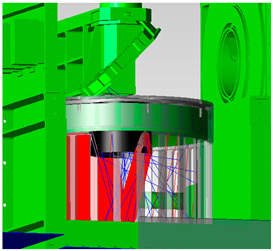

## Data Availability

Not applicable.

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
