# Peer review of "Analysis and Suppression Design of Stray Light Pollution in a Spectral Imager Loaded on a Polar-Orbiting Satellite"

_sensors, 2023, doi:10.3390/s23177625_

Round 1
Reviewer 1 Report
See comments in pdf attached

English quality is average, acceptable but not perfect. Some English language improvements are mentionned in the pdf
Reviewer 2 Report
A comprehensive stray light analysis method has been proposed, which first guides the design and simulation through a stray light suppression model, and then verifies the feasibility of the method through experiments and good results are obtained.However, there are still some minor problems,I recommend this paper to be publish after several minor revisions:
1. To facilitate readers' comprehension, the author should improve the quality of all the images.
2. In the formula section, the author should explain the meaning of each parameter in each equation.
3. The discussion on the relative radiation results only provides conclusions, and the previous and subsequent articles do not reflect the conclusions. The description is too simple, and it is recommended to enrich it.The article should explain how the proposed sensor works in practical applications.
4. Many of the charts in the article are not clearly reflected in the figures, lacking intuitiveness. For example, the PST value at 10 ° is not indicated in the figure.
5. For the evaluation of stray light suppression effect outside the field of view, there is a lack of validation in the description of the results, such as why a PST greater than 10 ° off axis is used to meet the requirements of stray light suppression to explain the stray light suppression effect, and the reason is not given.
A comprehensive stray light analysis method has been proposed, which first guides the design and simulation through a stray light suppression model, and then verifies the feasibility of the method through experiments and good results are obtained.However, there are still some minor problems,I recommend this paper to be publish after several minor revisions:
1. To facilitate readers' comprehension, the author should improve the quality of all the images.
2. In the formula section, the author should explain the meaning of each parameter in each equation.
3. The discussion on the relative radiation results only provides conclusions, and the previous and subsequent articles do not reflect the conclusions. The description is too simple, and it is recommended to enrich it.The article should explain how the proposed sensor works in practical applications.
4. Many of the charts in the article are not clearly reflected in the figures, lacking intuitiveness. For example, the PST value at 10 ° is not indicated in the figure.
5. For the evaluation of stray light suppression effect outside the field of view, there is a lack of validation in the description of the results, such as why a PST greater than 10 ° off axis is used to meet the requirements of stray light suppression to explain the stray light suppression effect, and the reason is not given.
Reviewer 3 Report
Stray light has an important impact on the normal imaging and data quantification applications. The so-called full link method was proposed and could be used for stray light suppression design for polar-orbit spectral imagers.
Comments:
1. It was claimed that the method for stray light suppression design for polar-orbit spectral imagers in the conclusion. Is it also suitable for the early-morning-orbit spectral imagers? Such as FY-3E MERSI-LL.
2. You said that the actual effect of stray light suppression had been verified through ground tests and on-orbit applications. where were the results of the on-orbit applications?
3. Line 167: What are the forward and reverse tracing analysis methods? These methods should be clearly stated.
4. Please check the reference in the full text.
